# Nested retrotransposition in the East Asian mouse genome causes the classical *nonagouti* mutation

Akira Tanave[1,3], Yuji Imai[1] & Tsuyoshi Koide [1,2]

Black coat color (nonagouti) is a widespread classical mutation in laboratory mouse strains. The intronic insertion of endogenous retrovirus VL30 in the *nonagouti* (*a*) allele of *agouti* gene was previously reported as the cause of the nonagouti phenotype. Here, we report agouti mouse strains from East Asia that carry the VL30 insertion, indicating that VL30 alone does not cause the nonagouti phenotype. We find that a rare type of endogenous retrovirus, β4, was integrated into the VL30 region at the *a* allele through nested retrotransposition, causing abnormal splicing. Targeted complete deletion of the β4 element restores *agouti* gene expression and agouti coat color, whereas deletion of β4 except for a single long terminal repeat results in black-and-tan coat color. Phylogenetic analyses show that the *a* allele and the β4 retrovirus originated from an East Asian mouse lineage most likely related to Japanese fancy mice. These findings reveal the causal mechanism and historic origin of the classical *nonagouti* mutation.

[1] Mouse Genomics Resource Laboratory, National Institute of Genetics, 1111 Yata, Mishima, Shizuoka 411-8540, Japan. [2] Department of Genetics, SOKENDAI (The Graduate University for Advanced Studies), 1111 Yata, Mishima, Shizuoka 411-8540, Japan. [3] Present address: Laboratory for Mouse Genetic Engineering, RIKEN Center for Biosystems Dynamics Research, 1–3 Yamadaoka, Suita, Osaka 565-0871, Japan. Correspondence and requests for materials should be addressed to T.K. (email: tkoide@nig.ac.jp)

Coat color mutations are common in laboratory animals. The reference laboratory mouse strain C57BL/6 (B6) carries the *nonagouti* (*a*) mutation at the *agouti* locus. The coloration of each hair in the *agouti* (*A*) mouse, the so-called wild type, shows unique hair pigmentation banded in a pattern of black-yellow-black from the root to the tip, whereas *nonagouti* (*a/a*) mice exhibit simple black hairs. From the early days of genetic studies after the rediscovery of Mendel's law, many researchers have been interested in the inheritance of the various alleles at the *agouti* locus, including the *a* allele, because of its ease of detection, high penetrance, and large number of available genetic resources[1,2]. More than 100 mutations at the *agouti* locus have been reported in the Mouse Genome Informatics (MGI) version 6.12 database, and the *a* allele is broadly distributed in a variety of inbred strains listed in the International Mouse Strain Resource (IMSR). These *nonagouti* mutants have been used for a variety of studies, such as analyses of the mechanism of pigmentation, melanosome function and its mechanism of gene regulation, and associations between coat color and behaviors[1,3–6]. Therefore, the *nonagouti* mutation remains useful in many research fields.

The *a* allele is characterized by a hypomorphic mutation of the *agouti* gene, which encodes a paracrine signaling molecule called agouti signaling protein (ASIP or ASP), and the reduced ASIP is not sufficient for switching between the synthesis of eumelanin (black or brown) pigment to that of pheomelanin (yellow) pigment. Thus, eumelanin is the predominant pigment in the melanocytes within the hair follicle in B6 mice[1,3]. The *agouti* gene, located on chromosome 2, is composed of seven exons. A previous genetic study proposed that an insertion of 11 kb in the intron lying downstream of the *agouti* exon 1C causes the non-agouti phenotype in *a/a* mice[4]. The structure of the inserted sequence is complex and was estimated to consist of 5.5 kb of virus-like 30S (VL30) sequence, belonging to one of the murine endogenous retroviruses, as well as incorporating 5.5 kb of additional unknown sequence internally. The VL30 insertion could be mutagenic to the *agouti* gene, as such sequences can act as a target of epigenetic silencing[7], but the actual molecular mechanism underlying the decline in *agouti* gene expression remains unknown. Furthermore, the molecular nature and evolutionary process responsible for the internal, unknown sequence within the VL30 element still remains to be clarified.

Here, we characterized the molecular structure of the 11 kb sequence inserted into the *agouti* gene. We found that the internal sequence is a rare type of endogenous retrovirus, β4, which was integrated into the VL30 sequence through nested retrotransposition. Deletion of the internal β4 in a nonagouti strain using the CRISPR/Cas9 method caused reversion of the wild-type agouti phenotype, whereas deletion of the β4 except for a single long terminal repeat (LTR) sequence resulted in a black-and-tan coat color. These results clearly demonstrate that the nested retrotransposition of the β4 is the true cause behind the classical nonagouti mutation. Furthermore, we found that the retrotransposition of β4 was present in Japanese fancy mice, through which it was likely introduced into many laboratory strains. This study provides a clear scenario concerning the origin of one of the most famous classical mouse mutations.

## Results

**Genomic structure of the *nonagouti* allele.** We first characterized in detail the structure of the inserted sequence in the *a* allele based on the B6 reference genome sequence (Fig. 1a–c). We found that the inserted sequence consists of 5357 bp of VL30 sequence in an antisense orientation relative to the direction of *agouti* gene transcription, interrupted by 9344 bp of

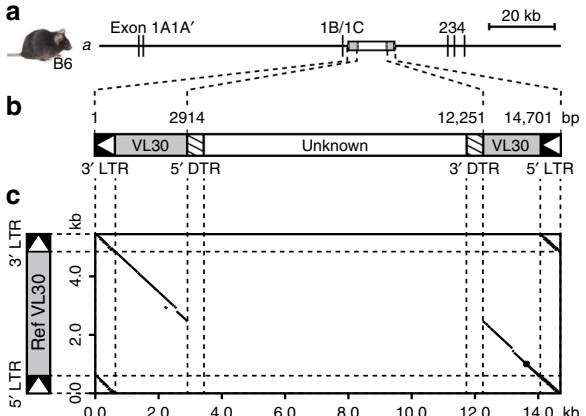

**Fig. 1** Structure of inserted sequence in the *a* allele. **a**, **b** Schematic representation of the genomic structure of the *a* allele of the *agouti* gene (**a**) and the inserted sequence (**b**) in the B6 mouse. The B6 reference genome sequence (14,705 bp in length on chr2:155,014,951–155,029,655 in GRCm38/mm10 assembly) was used to analyze the inserted sequence. **c** Comparison of the inserted sequence in the *a* allele (horizontal axis) and the reference VL30 sequence (X17124.1; vertical axis). Scale bar = 20 kb. LTR, long terminal repeat; DTR, direct terminal repeat

additional sequence (hereafter referred to as the unknown element) internally at position 2914 in the VL30 sequence. The length of the unknown element within VL30 was thus longer than that estimated in a previous report[4].

**Nested insertion of an endogenous retrovirus within VL30.** The presence of a 4-bp direct duplication, also called a target site duplication, flanking the VL30 element is thought to be evidence of a retroviral insertion in the *a* allele of the *agouti* gene (Fig. 2a)[4]. Similarly, we identified a 6-bp direct duplication of a part of the VL30 sequence on either side of the unknown element (Fig. 2a), implying that the unknown element was inserted into the VL30 sequence through a mechanism similar to the integration of retroviral sequence[8]. In support of this idea, we found a completely identical direct terminal repeat of 522 bp at both ends lying just inside the 6-bp direct duplication sequence. A sequence (GenBank ID: AB242436) homologous to the unknown element has been reported and derives from the inserted sequence in the *piebald* (*s*) allele of the endothelin receptor type B (*Ednrb*) gene (*Ednrb^s*) in Japanese fancy mouse strain JF1/Ms (JF1) (Fig. 2b)[9]. In order to predict the functional significance of the direct terminal repeats in the unknown element, we searched for homologous RNA sequences within the National Center for Biotechnology Information (NCBI) *Mus musculus* Annotation Release 106 using BLASTN (version 2.8.0 + ). Among these RNA sequences, we found eight genes that contain sequence homologous to the direct terminal repeat (*C6*, *Fuca1*, *Gimap3*, *Nyx* and *Serpina3a* mRNAs; *Mir680-1*, *Mir680-2*, and *Mir680-3* microRNAs). Based on available expression sequence tags and full-length cDNAs from mouse and experimental evidence from the *Ednrb^s* gene[8], we identified a transcription start site and a termination site on the sense strand of the homologous sequences (Supplementary Fig. 1). For instance, the direct terminal repeats in the unknown element were homologous to a 518-bp sequence composed of the first exon and the flanking downstream regions of the *Serpina3a* gene (93.1% identity), a 518-bp sequence composed of the last exon and the flanking upstream regions of the *C6* gene (87.4% identity), and a 507-bp sequence composed of the last exon and the flanking upstream regions of the abnormal transcript of the *Ednrb^s* gene (96.0% identity). It is known that the

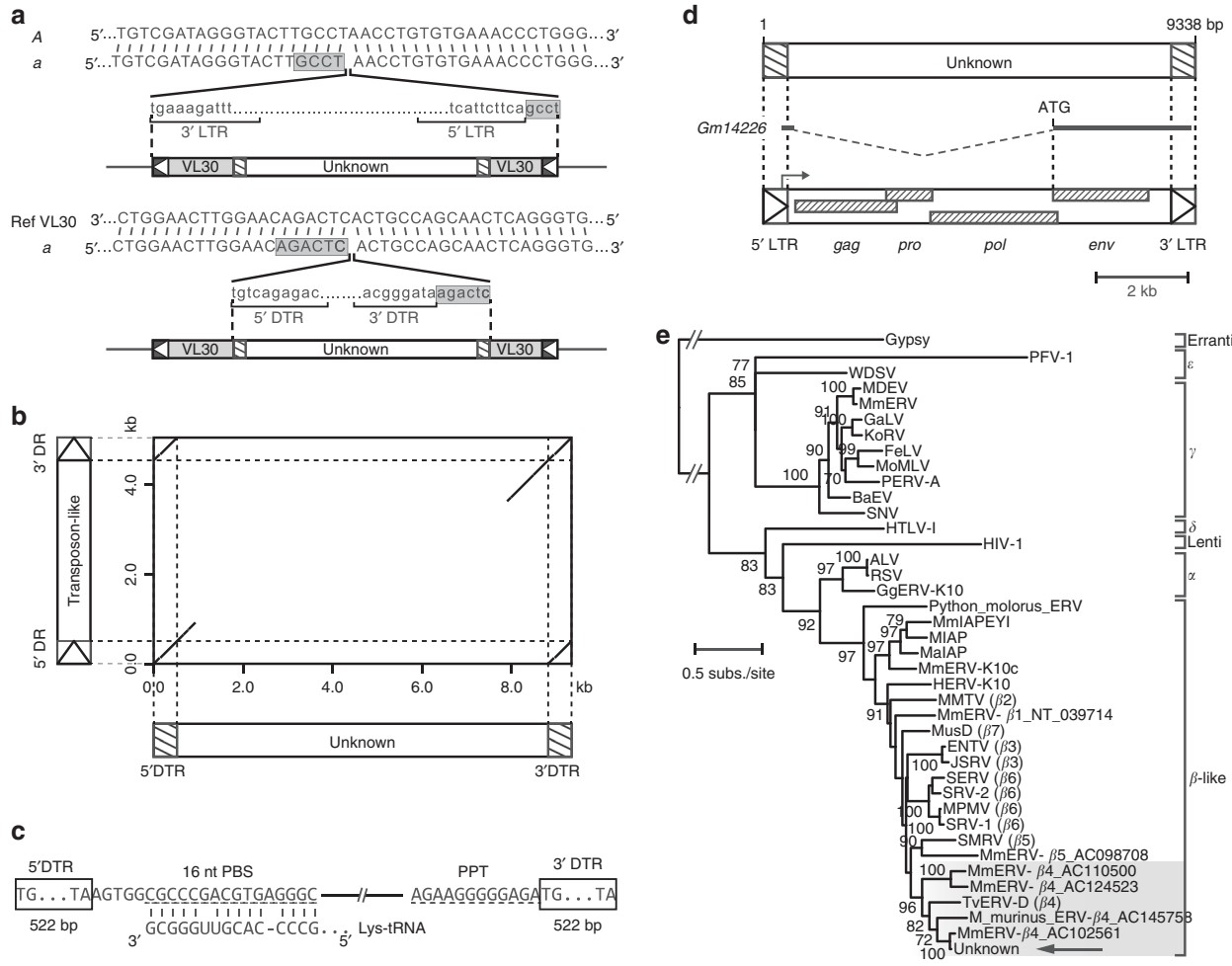

**Fig. 2** The unknown sequence inserted in VL30 in the *a* allele belongs to a member of the β4 endogenous retrovirus family. **a** Nucleotide sequence spanning the insertion site of the VL30 in the *a* allele compared to that of wild-type *agouti* (*A*) allele, and the unknown sequence within the VL30 compared to the reference VL30. The duplicated sequences are indicated by gray boxes. **b** Comparison of the inserted sequence in the *a* allele (horizontal axis) and the transposon-like sequence in the *Ednrb^s* gene (vertical axis). **c** Nucleotide sequences of the PBS and the PPT regions in the unknown sequence. Comparison of the sequence of Lys-tRNA against the PBS is shown below the PBS sequence. **d** Schematic representation of the *Gm14226* gene and the ORFs in the unknown sequence. ORF structures found in the unknown element by NCBI's ORFfinder (version 0.4.3) are indicated by stroked boxes. Scale bar = 2 kb. **e** Maximum likelihood neighbor-joining tree based on alignments of highly conserved regions of *pol* among major exogenous and endogenous retroviruses. The phylogenetic tree was rooted using Gypsy as the outgroup. Scale bar = 0.5 nucleotide substitutions per site. LTR, long terminal repeat; DTR, direct terminal repeat; DR, direct repeat

transcribed RNAs of retroviruses contain untranslated 5′ (U5), untranslated 3′ (U3,) and repeated (R) regions. Based on the transcription start and termination sites of the homologous genes (*Serpina3a*, *C6*, and *Ednrb^s*), we assumed that the sequences from the transcription termination site to the 3′ end (nt 426 to 522), from the 5′ end to the transcription start site (nt 1 to 403), and the repeated sequence (nt 404 to 425) in the direct terminal repeat of the unknown element corresponded to the retroviral U5, U3, and R regions, respectively (Supplementary Fig. 1). At the end of the U3 and upstream of the U5 regions, we identified a poly-adenylation signal (nt 328 to 332) and a TATA box (nt 396 to 401), respectively (Supplementary Fig. 1). Through comparison of the sequences in Fig. 2b, the homologous sequences were extended into adjacent internal regions between the 5′ and 3′ direct terminal repeats. Within these regions, we identified a primer binding site (PBS) that is complementary to a Lys-tRNA and a polypurine tract (PPT), which are necessary for replication and reverse transcription of LTR retrovirus RNA (Fig. 2c). These features indicate that the unknown element within the VL30 sequence is a retrovirus-like sequence.

Furthermore, current gene annotation in the MGI database indicates that the unknown element within the VL30 in the *a* allele covers the entire sequence of the *Gm14226* gene (predicted gene 14226), located in the sense orientation relative to the direction of *agouti* gene transcription (Fig. 2d). The *Gm14226* gene encodes a predicted protein of 686 amino acids, the function of which is unknown. To estimate the function of this predicted protein, we searched for homologous proteins using a BLASTP (version 2.8.0 +) search against annotated proteins. This search indicated that the overall amino acid sequence is 37–39% identical to the envelope proteins of murine and feline leukemia viruses (GenBank ID: AAP13891 and BAK41650, respectively). Retroviral sequences generally contain a set of viral genes, including *gag*, *pro*, *pol*, and *env*[10]. To determine whether the unknown element also encodes these genes, we analyzed the entire sequence of the unknown element in terms of its gene content. Open-reading frame (ORF) analysis revealed that the unknown element possesses intact *gag*, *pro*, and *pol* ORFs and an intact *env* ORF corresponding to the coding sequence of the *Gm14226* gene (Fig. 2d, Accession number: BR001522). This

further indicates that the unknown element within the VL30 sequence represents a full-length retrovirus sequence with identical 5′ and 3′ LTRs in the sense orientation relative to the direction of *agouti* gene transcription, implying the possibility that transcriptional read-through of the *agouti* gene may be affected by a cryptic splice acceptor site or the regulatory signals in the retroviral LTR.

**The unknown element inserted within VL30 is a β4 retrovirus.** In order to identify the retroviral family to which the unknown element belongs, we compared its *pol* sequence to those of both exogenous and endogenous retroviruses (Fig. 2e). Phylogenetic analysis revealed that the unknown element clusters together with the poorly characterized betaretrovirus group 4 (β4) of mouse endogenous retroviruses[11]. The genus *Betaretrovirus* includes type B and type D retroviruses, such as mouse mammary tumor virus (MMTV)[12] and *Mus musculus* type D retrovirus (MusD)[13]. The LTR and internal sequences of the unknown element showed 99.6% and 91.7% identities to the 522-bp LTR and 8316-bp internal sequences (ERVB4_1B-LTR_MM and ERVB4_1-I_MM, respectively) of the reference β4 (Supplementary Fig. 2). Notably, we noticed that the sequence of the unknown element is completely identical to that of the previously identified MmERV-β4_AL805955[11], a β4 element found in the AL805955 locus (mouse DNA sequence from clone RP23–192A6 on chromosome 2), and the locations of these elements are exactly the same (chr2:155,017,864 in GRCm38 assembly). These findings suggest that the *a* allele arose by nested insertional events of the endogenous retrovirus β4 into the VL30 in the *agouti* gene. Given that the name of MmERV-β4 may be easily confused with the previously identified *M. musculus* endogenous retrovirus (MmERV)[14], a type C retrovirus, we hereafter refer to the unknown element as β4.

**Agouti gene expression is disrupted by the β4 insertion.** We next investigated *agouti* gene expression to reveal the molecular mechanisms that disrupt the gene expression of the *a* allele. Similar to a previous report[4], available RNA-seq data[15,16] obtained from the skin of nonagouti B6 (*a/a* genotype) and wild-type agouti C3H/Kam (C3H, *A/A* genotype) mice showed that in B6, exon 1B/1C, but not exons 2–4, of the *agouti* transcript was expressed at levels comparable to that in C3H (Fig. 3a). Sashimi plot revealed that the splicing between exons 1B/1C and 2 was completely absent in B6 (Fig. 3a). Instead of the normal splicing, exon 1B/1C was spliced to either the flanking sites of the VL30 3′ LTR or the U5 region of the β4 5′ LTR in B6. Although it was difficult to precisely map the short reads onto the β4 LTR sequences at the 5′ and 3′ ends because of the complete identity between these sequences and related sequences in the mouse genome, we assumed that this abnormal transcript terminated at the β4 3′ LTR, corresponding to the structural terminal of the *Gm14226* gene. Reverse transcription PCR (RT-PCR) analysis of the *agouti* transcripts showed three major aberrant splicing products (transcripts 1, 2, and 3) in B6 but not in C3H and the normal *agouti* transcript (transcript 4) in C3H but not in B6 (Fig. 3b, c).

Several RNA-seq peaks were observed at both LTR and internal regions in the VL30 and β4 elements in B6 and C3H due to problems arising from mapping repetitive sequences to the mouse genome (Fig. 3a). Most of the paired-end reads mapped within each element and not beyond the element junctions, suggesting the possibility that these reads were derived from other genomic sequences or that there was genetic variation among inbred strains. However, > 100 spliced reads that included nucleotides from exons 1B/1 C to β4 (33 reads) and from the VL30 LTR to β4

(96 reads) showed clear evidence of the presence of the abnormal *agouti* transcript in B6.

To correct this abnormal splicing, we created mutants with deletions of the β4 or the VL30 + β4 sequences in the *agouti* gene by CRISPR/Cas9-mediated deletion, resulting in an intact VL30 or an intact *agouti* allele, respectively (Fig. 3d and Supplementary Fig. 3). Both the deletion alleles of β4 (β4-del) and of VL30 (VL30-del) showed recovery of the agouti coat color phenotype and normal splicing of the *agouti* transcript, indicating that the β4 insertion is the real cause of the nonagouti phenotype (Fig. 3e). Notably, the inclusion of a flanking sequence and part of the VL30 3′ LTR sequence in the 5′ untranslated region (UTR) of the *agouti* transcript (transcripts 5 and 6) was observed in β4-del mice (Fig. 3f).

**The solo β4 LTR within VL30 causes a black and tan phenotype.** Previously, it has been reported that excision of a large part of the unknown element (β4 sequence) in the *a* allele results in a phenotypic change from the nonagouti (*a*) to the black-and-tan (*a^t*) phenotype (Fig. 4a)[4]. In spite of these results, our CRISPR experiments showed restoration of the agouti coat color phenotype upon deletion of the complete β4 sequence (Fig. 3d and Supplementary Fig. 3d). In analyzing this discordance, we noticed that the *a^t* and the *a^t*-related alleles still carried a single LTR of β4 in VL30[4], and we postulated that this solo LTR was integral to expression of the black-and-tan phenotype. To test this hypothesis, we developed a *β4(direct-rec)* allele that still possessed the solo β4 LTR in the VL30 sequence by direct-repeat recombination of the 522-bp repeats of the LTR sequence in the β4 element in the *a* allele during the above CRISPR experiments (Fig. 4a and Supplementary Fig. 4a, b). As expected, *β4(direct-rec)* mice exhibited yellow coat color of the ventral but not the dorsal hairs (Fig. 4b). To evaluate *agouti* gene expression, we studied the *agouti* transcripts expressed from four alternative promoters: the ventral-specific exon 1A/1A′ and hair cycle-specific exon 1B/1C[4] in the neonatal ventral and dorsal skins (Fig. 4c). RT-PCR analysis showed that *β4(direct-rec)* mice expressed *agouti* transcripts only from exon 1A/1A′ (transcripts 8 and 9, also called form II transcripts; and transcript 7) in the ventral skin, consistent with a previous study[4]. In both the dorsal and ventral skins from β4-del and VL30-del mice, we observed several transcripts expressed from exons 1A/1A′ and 1B/1C that contained the VL30 LTR (transcripts 7, 5, and 6) or intronic sequences (transcript 11), in addition to normal transcripts (transcripts 8, 9, 10, and 4), which exhibited proper splicing, including the protein-coding exons 2–4. Abnormal transcripts expressed from exons 1A/1A′ and 1B/1C were also observed in both the dorsal and ventral skins of homozygous *β4(direct-rec)* mice (transcripts 7 and 13), although this was clearly less common than in homozygous nonagouti mice in the ventral skin (Fig. 4d and Supplementary Fig. 5). These results demonstrated that the black-and-tan phenotype is caused by a single β4 LTR sequence that completely disrupts expression of the *agouti* transcript from exon 1B/1C in both the dorsal and ventral skins but does not disrupt expression of the *agouti* transcript from exons 1A/1A′ in the ventral skin (Fig. 4d).

**The VL30 in the nonagouti allele was derived from wild mice.** Next, we analyzed the evolutionary relationships among the alleles of the *agouti* locus in inbred laboratory and wild mouse strains to determine the historic origin of the *a* allele. Wild mice, *M. musculus*, are classified into three representative subspecies groups, *domesticus*, *musculus*, and *castaneus*, based on genetic characteristics. These subspecies groups are further subdivided into many taxonomic subspecies. For example, the Japanese mouse, *M. m. molossinus*, is categorized as a subspecies belonging

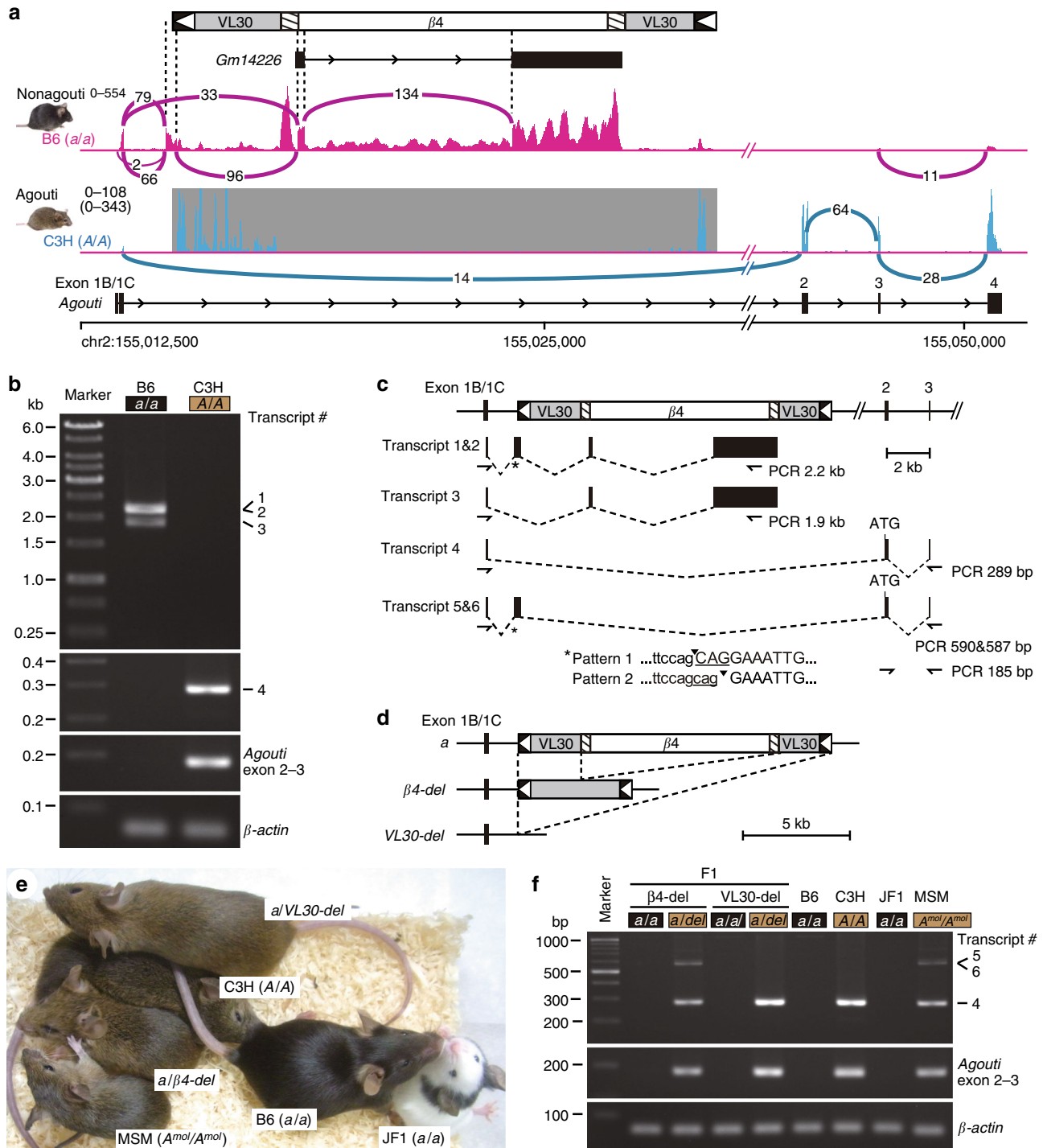

**Fig. 3** The β4 insertion eliminates *agouti* gene expression. **a** Sashimi plot of RNA-seq reads of the *Agouti* gene. Data from dorsal skin RNAs[15] from a neonatal B6 (nonagouti; *a/a*) mouse (magenta) and skin RNAs[16] from a C3H (agouti; *A/A*) mouse (blue) were mapped onto the B6 genome sequence (magenta horizontal bar). Arcs connecting reads and their respective read numbers represent junctional reads aligned with the gap. Genomic region absent in the C3H strain is shown by a gray background, and the maximum of the read depths, including the absent region is shown in parentheses. **b** RT-PCR of the *agouti* transcripts in dorsal skins of a neonatal B6 (*a/a*) mouse. A C3H mouse (*A/A*) was used as a positive control. Transcript numbers are shown on the right. **c** *Agouti* transcript structures detected via RT-PCR analyses. *The splice junction is indicated by a filled triangle and the flanking sequence by lowercase letters in the nucleotide sequence. Scale bar = 2 kb. Primers used for RT-PCR are marked by arrows. **d**, **e** Coat color phenotypes and schematic representation of the genomic structure of the *agouti* locus in adult B6 (*a/a*), β4-del (*β4-del*), and VL30-del (*VL30-del*) mice. **d** Deletions of the β4 or the VL30, including the β4 in the *a* allele were mediated by the CRISPR/Cas9 system. Scale bar = 5 kb. **e** Representative appearance of the deletion mutants. Heterozygous β4-del (*a/β4-del*) and heterozygous VL30-del (*a/VL30-del*) mice showed agouti coat colors. Wild-type agouti (C3H; *A/A*), wild-type white-bellied agouti (MSM; *A^mol/A^mol*), nonagouti (B6; *a/a*), and nonagouti with piebald (JF1; *a/a*) mice are shown as positive or negative controls. **f** RT-PCR of the *agouti* transcripts in neonatal dorsal skins of heterozygous β4-del (*a/β4-del*) and heterozygous VL30-del (*a/VL30-del*) mice. Nonagouti F1 littermates (*a/a*) were used as negative controls. *β-actin* was used as an internal control. Scale bar = 200 bp

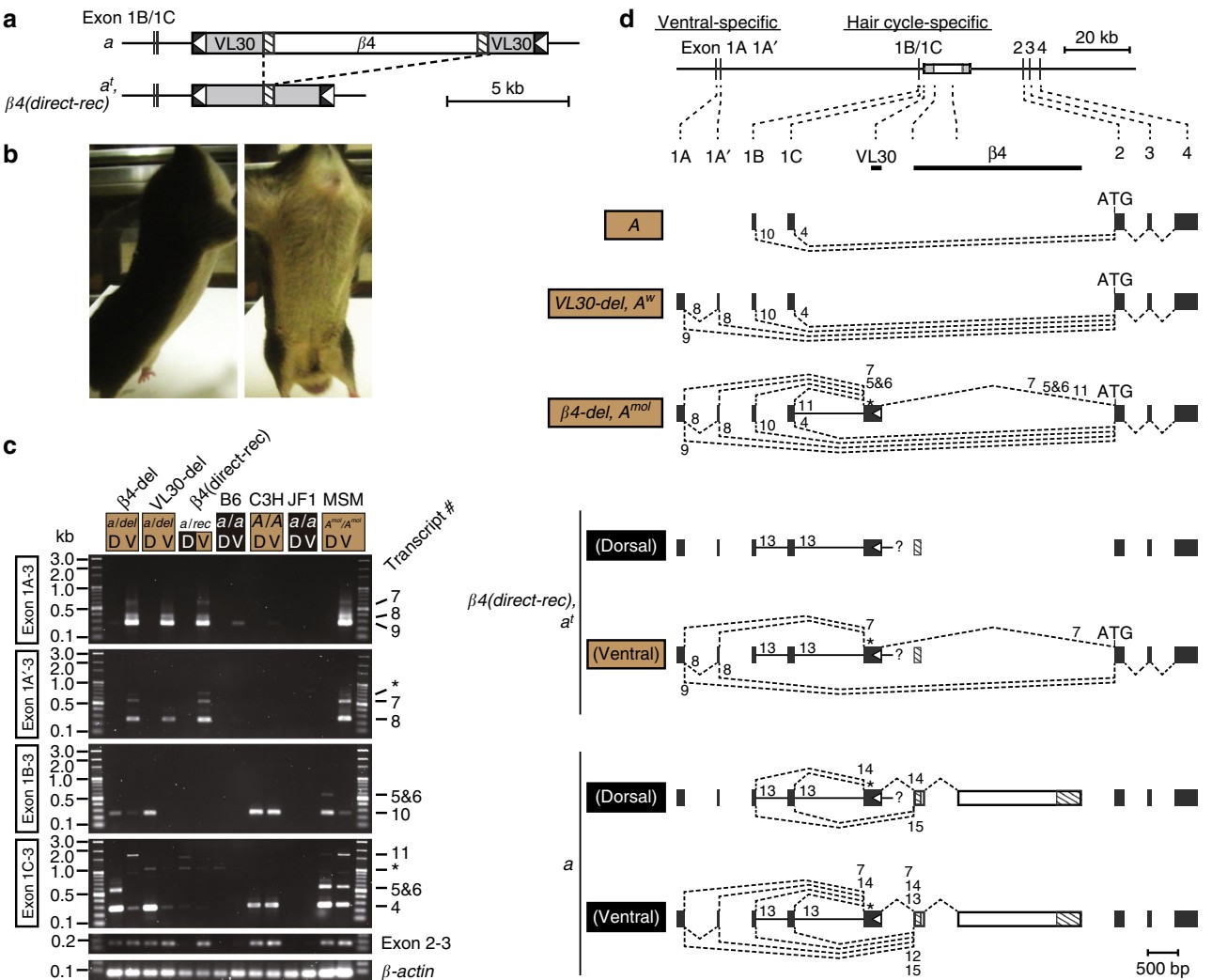

**Fig. 4** The solo β4 LTR sequence interrupts *agouti* gene expression in dorsal but not ventral skin. **a** Schematic of homologous recombination from *nonagouti* (*a*) to *black-and-tan* (*a^t*) or *β4(direct-rec)* alleles. **b** Representative appearance of the β4(direct-rec) mutant. Heterozygous β4(direct-rec) (*a/β4(direct-rec)*) mice exhibited yellow color in ventral but not dorsal hairs. **c** RT-PCR of the *agouti* transcripts in dorsal and ventral skins of neonatal β4(direct-rec) (*a/β4(direct-rec)*), and β4-del (*a/β4-del*) mice. The *agouti* transcripts were amplified from exons 1A, 1A', 1B, and 1C to 3. Transcript numbers are shown on the right. The *agouti* transcripts from exons 2 to 3 were also amplified. *β-actin* was used as an internal control. Asterisks indicate not analyzed. Wild-type agouti (C3H; *A/A*), wild-type white-bellied agouti (MSM; *A^mol/A^mol*), nonagouti (B6; *a/a*), and nonagouti with piebald (JF1; *a/a*) mice are shown as positive or negative controls. D, dorsal skin; V, ventral skin. **d** *Agouti* transcript structures detected via RT-PCR analyses. Transcript numbers are shown beside the splicing connections. The splicing patterns of dorsal and ventral skins are shown separately for each β4(direct-rec) and nonagouti (*a*) allele. Scale bar = 500 bp

to the *musculus* group[17,18]. A phylogenetic tree based on an alignment of the full-length sequences of the *agouti* gene (Fig. 5a) in various inbred strains using next-generation sequencing (NGS) reads[19,20] showed that the nonagouti laboratory strains clustered into the same group with Japanese fancy mouse strain JF1 and other East Asian *musculus* group strains, including CHD/Ms (CHD), SWN/Ms (SWN), KJR/Ms (KJR), MOLF/EiJ (MOLF), and MSM/Ms (MSM). This cluster contained laboratory strains that carry the *a* allele at the *agouti* locus. Other laboratory strains showing the agouti phenotype were clustered together with *domesticus* group wild strains ZALENDE/EiJ, BFM/2Ms, PGN2/Ms, WSB/EiJ (WSB), and LEWES/EiJ, consistent with previous reports[21,22]. As most parts of the genomes of laboratory strains are derived from the *domesticus* group[23], the clustering indicated an introgression of the original *a* allele in the lineage of the East Asian *musculus* group, including JF1 into the laboratory strains.

We next analyzed the structural variation of the *agouti* locus in inbred mouse strains (Fig. 5a and Supplementary Fig. 6). NGS

reads[19,20] showed that a majority of classical laboratory strains with the *a/a* genotype and JF1 have both VL30 and β4 insertions, whereas wild strains MOLF and MSM have only the VL30 insertion at the *agouti* locus. Importantly, however, there was no paired-end read aligned over both sides of the flanking sequence of VL30 at the *agouti* locus in MOLF and MSM mice, indicating that the inserted site is interrupted by a sequence longer than the fragmented DNA size (200 bp on average) in the NGS data. Capillary sequencing data[20] also showed the presence of both ends of VL30 at the *agouti* locus in MSM (Supplementary Fig. 7). We verified this insertion by genomic PCR of the *agouti* gene at the inserted site (Fig. 5b). PCR products in strains with the *a/a* genotype were about 15 kb in length, corresponding to the full-length VL30 and β4 insertions in the *a* allele, while those of CHD, KJR, SWN, and MSM were about 5.6 kb in length, corresponding to the full-length VL30 insertion only. Direct sequencing of the PCR products revealed that the *agouti* locus of CHD, KJR, SWN, and

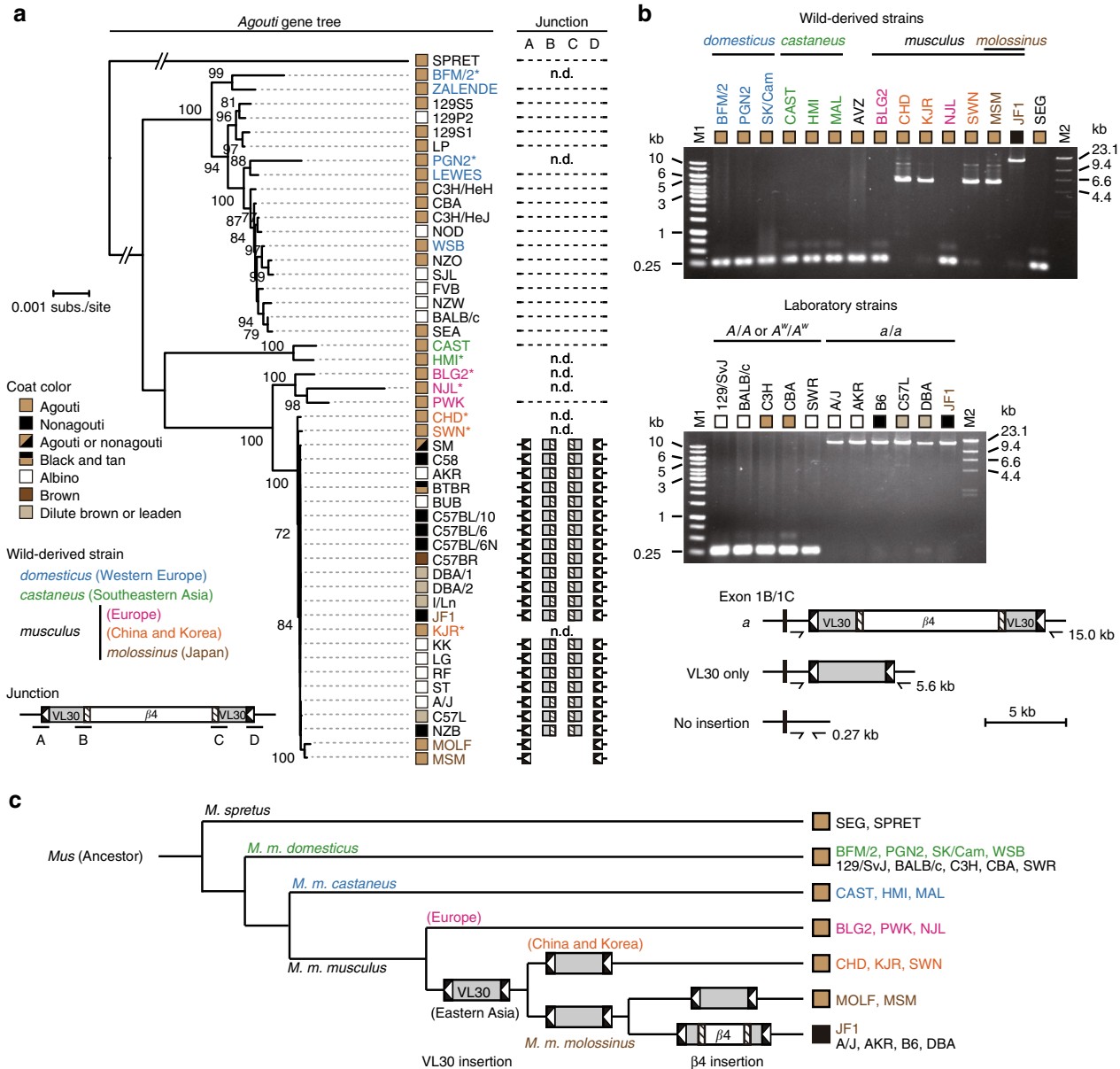

**Fig. 5** Two insertional mutation events occurred at the *agouti* locus during the differentiation and domestication of the Japanese mouse. **a** Phylogenetic relationships of the structural variations in the *agouti* locus in various inbred strains. The absence of the insertion is indicated by a connected line. The maximum likelihood neighbor-joining tree was constructed based on alignments of the full-length sequence of the *agouti* gene (chr2:154,951,180–155,051,012). Bootstrap values are shown as a percentage of 10,000 replicates, and only bootstrap values above 70% are shown. The *M. spretus*-derived SPRET/EiJ (SPRET) was used as an outgroup. Scale bar = 0.001 nucleotide substitutions per site. n.d., no data available. **b** Gel-electrophoresis of the PCR-amplified DNA fragments spanning the insertion site of VL30 in the *agouti* loci of various wild-derived strains and laboratory strains. An *M. spretus*-derived SEG/Pas (SEG) mouse was used as an outgroup. A schematic illustration of the structures of the amplified DNA fragments and their respective amplicon lengths are shown on the bottom. M1; 1 kb marker. M2; Lambda DNA/*Hind*III marker. Scale bar = 5 kb. **c** Schematic illustration of the putative evolutionary and historic transition of the *agouti* alleles

MSM contains an intact VL30 insertion that is not interrupted by the β4 insertion (Supplementary Fig. 8).

To evaluate *agouti* gene expression in these wild mice that contain an intact VL30 insertion, we performed RT-PCR analysis. As observed in β4-del mice, the MSM strain also showed the inclusion of a flanking sequence and part of the VL30 3′ LTR sequence into the 5′ UTR of the *agouti* transcript (Fig. 3f and Supplementary Fig. 5). MSM is an inbred strain derived from Japanese wild mice. The coat color phenotype of MSM mice is that of a white-bellied agouti and reported as the $A^w$ allele (see MGI database). Since *agouti* alleles of some wild mice are distinct

from those carried by laboratory mice, as mentioned in a previous report[21,22] and based on the results reported here, we refer to the *agouti* locus allele carried by MSM as $A^{mol}$.

Finally, we observed that the read-depth of β4 is 1 or 0 in the wild-derived *domesticus* group strains but >3 in Japanese fancy strain JF1 (Supplementary Figs. 9–11), implying that the copy-number of β4 is higher in this strain.

Our findings support the idea that the VL30 and the β4 sequences in the *a* allele were derived from step-by-step insertional events into an intact *agouti* allele in the lineage encompassing the East Asian *musculus* group and JF1 (Fig. 5c). In

this scenario, the VL30 endogenous retrovirus transposed into an intact wild-type *agouti* allele in the wild mouse population of the *musculus* group. The β4 endogenous retrovirus then transposed into the VL30 sequence in the lineage that separates JF1 from other East Asian wild-mice belonging to the *musculus* group. This second insertional mutation likely occurred relatively recently, based on the perfect identities of both the 6-bp direct duplication and the 5′ and 3′ LTR sequences of the β4 in the *a* allele.

## Discussion

Little is known regarding the genes or phenotypes that were intentionally introduced into classical laboratory mice from Japanese fancy mice. Limited knowledge of the gene flow of mutated alleles is available for the *pink-eyed dilution* (*p*) and *piebald* (*s*) genes that originally arose in Japanese mice[24,25]. The *a* allele has been studied intensively in laboratory mice and is common among the classical laboratory strains[26,27], but the genetic history that links the *nonagouti* mutation back to the wild-type *agouti* allele was previously obscured. Along with a previous study[22], our phylogenetic analyses indicate the same origin of the *a* allele in JF1 and other nonagouti laboratory strains, suggesting that the *nonagouti* mutation derived from a colony of fancy mice in Japan[25]. This presumable scenario is consistent with the fact that classical laboratory strains were developed by crossing Japanese and European fancy mice[20,23]. Our results also indicate that the *nonagouti* mutation, involving the nested insertion of β4 into VL30, occurred in the lineage, leading to the East Asian *musculus* subspecies group and *M. m. molossinus* before developing a founder stock of JF1.

Many phenotypic mutations caused by the insertion of an endogenous retrovirus have been reported in mouse[28,29]. Murine leukemia virus (MLV) and intracistemal A particle (IAP) are typical endogenous retroviruses that retain their transposable activity[30]. In the case of β4, its transposable activity in the mouse genome has not been studied further since it was identified[11]. In the present study, we found that the *nonagouti* (*a*) and the *piebald* (*s*) mutations were caused by the insertion of a full-length and truncated β4 in Japanese fancy mice, respectively. This provides good evidence that these mutations arose from infection with β4 exogenous retrovirus or from the retrotransposition of a β4 endogenous retrovirus at another locus. In the latter possibility, identification of a candidate locus with a sequence highly similar to that of the β4 in the *a* allele should be feasible through additional NGS studies in wild mice.

The impact of retroviral elements in the mammalian genome on gene expression has been of long-standing interest[29,31]. As a vast number of retroviral elements are located within or near coding, promoter, or enhancer sequences, they can affect gene expression through various mechanisms[32]. The *a* allele is an good example of a deleterious effect caused by proximal interference of complex retroviral elements with gene expression. In terms of molecular mechanisms, elements can affect splicing, transcription termination, and transcription levels of genes. Our results demonstrated that the full-length β4 sequence interrupts all transcription from the upstream region, the solo β4 LTR sequence only interrupts proximal transcription from exons 1B/1C, and the VL30 sequence oriented in the opposite direction can act as an alternative exon but does not interrupt transcription. We estimate that interference depends on the composition and complex structure of splicing acceptor sites, transcription termination signals, and other various epigenetic modifications such as DNA methylation, histone modification, and DNA binding proteins. For example, we noticed that part of the β4 internal sequence is highly enriched for sequences associated with the CCCTC-binding factor (CTCF) protein, which acts as an insulator or

anchor for chromatin loops[33]. Moreover, the β4 LTR sequence can be a target of epigenetic modification, suppressing proximal exon 1B/1C expression[34]. Further experiments into these epigenetic aspects are needed to elucidate the details of the molecular mechanisms associated with the retroviral elements.

For decades, it has been considered that the VL30 insertion is the cause of the nonagouti and black-and-tan phenotypes, as wild-type agouti alleles (*A* and *A^w*) do not contain the VL30 sequence[4,29,35,36]. However, the present study clearly shows that the *A^mol* allele, which carries VL30 without the unknown element (β4 sequence) is a true original wild-type allele for *nonagouti*, resulting in a white-bellied agouti phenotype. This finding clearly demonstrates that there is no association between VL30 itself and black coat color. In addition, a previous study reported that the black-and-tan phenotype is caused by excision of the unknown element (β4 sequence) by direct-repeat recombination of the 526-bp repeats located at both ends of the unknown element in the *nonagouti* allele, which contains both VL30 and β4[4]. Our present study clearly shows that nonagouti coat color is caused by the full-length β4 sequence, whereas the black-and-tan phenotype is caused by the solo LTR of the β4 in the *agouti* gene. However, our results do not exclude the possibility of the involvement of the VL30 sequence in altering the expression of the *agouti* gene. In particular, the black-and-tan phenotype may be caused by a combination of effects derived by VL30 and the solo β4 LTR. Even so, this study is the first report showing that the VL30 insertion in the *a* allele is not directly associated with the nonagouti phenotype and the reduced expression of the *agouti* gene.

VL30 retains its transposable activity under some artificial conditions[37,38], but the *a* allele is the only published evidence of a recent spontaneous mutation caused by a VL30 insertion. As VL30 is a non-autonomous retrovirus that lacks the retroviral genes required for transposition using its own mRNA[39], this discordance suggests that the transposable activity of VL30 in the germline is weaker than previously thought[40] and that it may no longer be active in the mouse genome under the normal conditions of animal breeding.

In contrast, the current mouse genome shows that the β4 endogenous retrovirus has been incorporated into several genes, as shown in Supplementary Fig. 1. This implies that the significance of β4 insertions is similar to that of other endogenous retroviruses that are thought to contribute to genetic variations, phenotypic changes, and evolution[10,41,42]. Consistent with the expression pattern of the *Serpina3a* gene, which is specifically expressed from the β4 LTR sequence in Sertoli cells[43], RNA-seq data[44], and microarray data[45] show the predominant expression of the *Gm14226* gene in the testis (Supplementary Fig. 12). This specific promoter activity in the host genome is similar to that of other endogenous retroviruses[46]. For example, we found that both the *nonagouti* (*a*) and the *piebald* (*s*) mutations in Japanese fancy mice are related to the β4 insertion. This co-occurrence in one lineage indicates the contribution of β4 to phenotypic variation, especially in terms of coat color mutations, during the domestication process of Japanese mice. Several historic literary sources from the Edo period describe the presence of black mice in Japan. Indeed, Ukiyo-e, an ancient Japanese artwork by Suzuki Harunobu ca. 1767–1768 shows a black mouse on a young man's hand (Fig. 6; Museum of Fine Arts, Boston).

Taken together, our data elucidate the molecular basis and history of the *nonagouti* mutation. Detailed characterization of the inserted sequence in the *a* allele revealed that the abnormal splicing of *agouti* transcripts results in the nonagouti phenotype. Until now, it was believed that insertion of the VL30 endogenous retrovirus in the *agouti* locus was the cause of the *nonagouti* mutation; however, this work demonstrates that this is not the

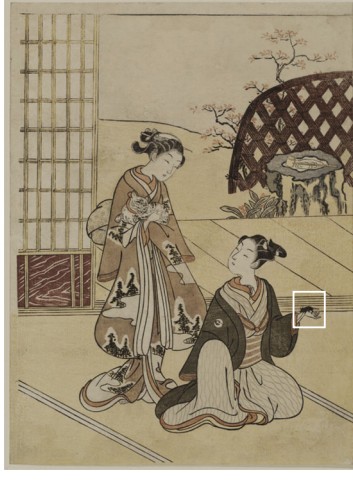
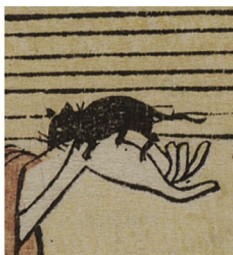

**Fig. 6** An example of an old Japanese literary source showing a black mouse. Title of the artwork is "Young Woman Holding Cat and Young Man Holding Mouse". Enlarged image of the region in the white box in the original panel is shown. Photograph © Museum of Fine Arts, Boston

case and that instead, insertion of the β4 endogenous retrovirus into the VL30 sequence is the actual cause of the *nonagouti* mutation. We propose that the *a* allele was derived via step-by-step insertional mutation events in the lineage leading from East Asian mice to Japanese fancy mice. This insertional mutation event of the rarely characterized endogenous retrovirus β4 is the origin of one of the most popular classical genetic mutant alleles in laboratory mice.

## Methods

**Genome sequence**. For structural analysis of the VL30 sequence and the unknown sequence in the *a* allele, the reference mouse genome sequence of the B6 strain (GRCm38/mm10) was obtained from the UCSC database.

**Dot plot analysis**. For visualization of similarity between two sequences, dot-plots were generated using PipMaker[47] (perl version: 2003-Jul-08) using the default parameter values. The reference VL30 sequence (X17124.1) and the transposon-like sequence in the *Ednrb^s* allele (AB242436.1) were obtained from NCBI.

**Phylogenetic analysis of retroelements**. The publicly available nucleotide sequences of various representative exogenous and endogenous retroviruses were obtained from NCBI or Repbase (Supplementary Data 1). The FASTA sequences were aligned with the Clustal Omega algorithm using the R package msa version 1.12.0 implemented in R version 3.4.3. The aligned sequences were constructed using highly conserved regions of *pol* nucleotides as described previously[11]. Briefly, the eight conserved regions of the polymerization domain of retroviral reverse transcriptase (corresponding to amino acid residues 23–266 of HIV-1 reverse transcriptase) were used for the alignment. Detailed information for each *pol* nucleotide sequence is provided in Supplementary Data 1. The distance matrix was constructed by the dist.ml function using R package phangorn version 2.4.0. A maximum likelihood neighbor-joining tree was constructed by the nj function using R package ape version 5.1. In order to fit models using the maximum likelihood tree, the best model was selected and applied in the tree using the modelTest, pml, and optim.pml functions in R package phangorn. The robustness of each branch was confirmed by bootstrapping analysis with 10,000 bootstrap samples using R package phangorn.

**RNAseq data analysis**. Publicly available FASTQ reads from the dorsal skin RNA of neonatal P4 B6 (ERR231641)[15] and adult C3H (SRR2886933)[16] mice were obtained from the EMBL-EBI database. The FASTQ reads were aligned to the reference genome sequence (GRCm38) using HISAT2[48] version 2.1.0 with SNPs and transcript indexes obtained from the ftp site of HISAT2. We applied the options '–dta–no-discordant–no-mixed' to the HISAT2 run. The mapped SAM data were compressed and sorted into a BAM file using SAMtools[49] version 1.5 (using htslib 1.5), and then the data were indexed. Sashimi plot was constructed by Integrative Genomics Viewer (IGV)[50] version 2.3. Representative mapping illustration was also constructed by IGV.

**Animals and sampling**. C57BL/6JJcl (B6), Jcl:ICR (ICR), and C3H/HeNJcl (C3H) mice were purchased from CLEA Japan (Tokyo, Japan). MSM/Ms (MSM), JF1/Ms (JF1), and JF1-*Ednrb^s+*/Ms (JF1-*s +*) mice were established at the National Institute of Genetics (NIG) (Mishima, Japan)[25,51]. The animals were bred in standard-sized plastic cages on wood chips at the NIG. The animals were maintained under standard environmental conditions (12-h light/dark cycle, light from 6 a.m. to 6 p.m.; temperature-controlled room, 23 ± 2 °C). Food and water were available ad libitum. The animals were weaned from 25 to 31 days of age and housed in same-sex groups. The animals were maintained according to NIG guidelines, and all experimental protocols were approved by the Institutional Animal Care and Use Committee at the NIG (approval identification number: 30–11).

**RNA extraction and cDNA synthesis followed by RT-PCR**. Neonatal P5 dorsal skins in each strain were used to quantify mRNA levels. All samples were collected between 2 p.m. and 6 p.m. The animals were sacrificed by decapitation, and then dorsal skins were sampled. Each tissue was homogenized in TRIzol™ reagent (15596026, Thermo Fisher Scientific, Waltham, MA, USA), and total RNAs were isolated from the homogenates according to the manufacturer's protocol followed by DNA digestion with a TURBO DNA-free™ Kit (AM1907, Thermo Fisher Scientific). First-strand cDNA was synthesized with a PrimeScript™ RT reagent Kit (Perfect Real Time) (6210 A, Takara Bio, Shiga, Japan) in accordance with the manufacturer's instructions using 0.5 μg of total RNA with oligo (dT) primer and random hexamers. RT-PCR was carried out with KOD FX Neo DNA polymerase (KFX-201, TOYOBO, Osaka, Japan) on a Veriti Thermal Cycler (4375786, Thermo Fisher Scientific). Gene-specific primers (Supplementary Table 1) were designed using Primer3[52] and Primer-BLAST[53]. The primers were checked to avoid mismatching against the MSM genome or transcripts using the NIG Mouse Genome Database (NIG_MoG)[54]. The mRNA levels were compared more than three times for each test. *β-actin* cDNA was amplified as a positive control. Original uncropped images are shown in Supplementary Fig. 13.

**Genome editing**. To design specific and efficient guide RNA sequences for the CRISPR/Cas9 system, the CRISPR Design Tool[55], CRISPRdirect[56], and CRISPOR[57] were used. Using the pX330 vector (#42230, Addgene, Watertown, MA, USA), template DNAs for in vitro transcription of the guide RNAs were amplified by PCR using KOD Neo DNA polymerase (KFX-401, TOYOBO) with a primer (Supplementary Table 1) containing the sequences of the T7 promoter and the target site. The primers were designed to mimic the previously reported optimized sgRNA sequence[58]. The guide RNAs were synthesized with the MEGAshortscript™ T7 Transcription Kit (AM1354, Thermo Fisher Scientific).

For preparing *Cas9* mRNA, linearized Cas9-poly(A) plasmid DNA[59], kindly provided by K. Yoshimi, was used for in vitro transcription using a mMESSAGE mMACHINE™ T7 ULTRA Transcription Kit (AM1345, Thermo Fisher Scientific) according to the manufacturer's protocol. The mRNA product was purified using a MEGAclear™ Transcription Clean-Up Kit (AM1908, Thermo Fisher Scientific).

For deletion of β4 or VL30 in the *a* allele, the guide RNA (50 ng/μl) and either *Cas9* mRNA (50 ng/μl) or Cas9 protein (1074181, Integrated DNA Technologies, Coralville, IA, USA) (50 ng/μl) were microinjected into the pronuclei of fertilized B6 oocytes with oligo DNA (20 ng/μl) designed to join both sides of the inserted site (Supplementary Fig. 3a). The direct duplication on both sides of the inserted endogenous retrovirus was corrected to mimic that of the original sequence (Supplementary Table 1). The 5′ and 3′ homologous arms were about 40 bp in length each. The injected zygotes were transferred into pseudopregnant female ICR mice. The pups were genotyped by genomic PCR (Supplementary Fig. 3b). A correctly targeted allele was confirmed by direct sequencing of the PCR product (Supplementary Fig. 3c). The β4-del and VL30-del heterozygous mice showed an agouti dorsum and cream-colored ventrum, similar to the coat color of spontaneous revertant of the *a* allele found in a B6 colony[1] (Supplementary Fig. 3d).

The β4(direct-rec) mice were obtained during the above CRISPR-mediated deletion of β4 in the *a* allele, possibly due to the double strand break at the end of β4. Direct-repeat recombination was confirmed by genomic PCR (Supplementary Fig. 4b) and direct sequencing of the PCR product (Supplementary Fig. 4a).

**Genomic PCR**. The sequence and the genomic positions of primers used for genomic PCR are shown in Supplementary Table 1. For long-range PCR, the DNA was amplified by step-down thermal cycling (one cycle of 98 °C for 2 min; three cycles of 98 °C for 10 s and 74 °C for 15 min; five cycles of 98 °C for 10 s and 72 °C for 15 min; seven cycles of 98 °C for 10 s and 70 °C for 15 min; 20 cycles of 98 °C for 10 s, and 68 °C for 15 min; one cycle of 68 °C for 8 min) using primers cartridge-purified with KOD FX Neo on a Veriti Thermal Cycler. Original uncropped images are shown in Supplementary Fig. 13.

**NGS data analysis**. The publicly available aligned BAM files of various inbred strains (Supplementary Table 2) and the reference genome sequence (GRCm38: Ensembl release 68) were obtained from the ftp site of the Mouse Genomes Project[19] of the Wellcome Sanger Institute. For the MSM strain, the publicly available FASTQ reads[20] (DRR000458-DRR000462) were obtained from the ftp site of the

DDBJ Center. Detailed information for each strain is provided in Supplementary Table 2.

For the alignment, the FASTQ reads of MSM were aligned to the reference genome sequence using BWA[60] version 0.7.16a-r1181 with options "-t 8 -M -R". The mapped SAM data were compressed, merged, and sorted into a BAM file using SAMtools, and then the data were indexed. Using Picard version 2.9.4 (http://broadinstitute.github.io/picard/), paired-end information was verified and fixed using the FixMateInformation command, and then duplicated reads were removed with the MarkDuplicates command. After indexing by SAMtools, the reads were realigned and recalibrated by the RealignerTargetCreator/IndelRealigner/BaseRecalibrator/PrintReads commands using the Genome Analysis ToolKit (GATK)[61] v3.8–0-ge9d806836 with dbSNP and indel information obtained from the ftp site of the Mouse Genomes Project. Finally, the aligned BAM reads were indexed by SAMtools.

To identify structural variants, the depths of the both sides of each endogenous retrovirus were analyzed. Deletion was annotated as a depth within the 5% tail of coverage for all autosomal sequences masked by repeatMaskter obtained from UCSC. The coverage was calculated by the DepthOfCoverage command using GATK with options '–minMappingQuality 20–minBaseQuality 0–countType COUNT_FRAGMENTS -rf BadMate -rf MateSameStrand–omitLocusTable–omitIntervalStatistics–omitDepthOutputAtEachBase'.

For constructing the phylogenic tree, the genomic sequences (GRCm38 assembly) excluding the repetitive elements in various inbred strains were determined using the BAM reads. For the whole genome, we used 100 bp of sequence at each of 4000 loci that were randomly chosen from autosomal chromosomes. The scalable variants were called with the HaplotypeCaller command using GATK with option '-ERC GVCF' and with dbSNP and indel information. All of the gVCF files were merged by the CombineGVCFs command using GATK, and then SNPs and indels were called by the GenotypeGVCFs command using GATK with dbSNP and indel information. This variant information was recalibrated by the VariantRecalibrator/ApplyRecalibration commands by GATK with mode SNP and INDEL, respectively. In some wild-derived strains (BFM/2, PGN2, HMI, KJR, SWN, CHD, NJL, BGL2, MSM, and JF1), the SNP data were obtained using the VariantTable function in the NIG Mouse Genome Database (NIG_MoG2)[54]. Applying only the homologous SNP variants, FASTA sequences for each strain were constructed by the FastaAlternateReferenceMaker command using GATK, in which repetitive sequences were excluded based on UCSC's RepeatMasker information. Finally, 141, 1368, and 11,889 SNPs were obtained in the 5-kb flanking sequences of the inserted VL30 (24,701 bp in length on chr2:154,951,180–155,051,012), the full-length sequence of the *agouti* gene (99,833 bp in length on chr2:154,951,180–155,051,012), and all autosomal chromosomes, respectively. The phylogenetic tree was constructed in a method similar to the analysis of retroelements, but the sequence alignment using the msa function in R was not applied because we only used SNP data in this analysis. The sequence of the SPRET strain was used as an outgroup.

**Haplotype calling of the β4 endogenous retrovirus**. A low copy-number region of the β4 sequence in the *a* allele was determined based on the abundance of the homologous sequence. In order to quantify the abundance, BLAST hits for the LTR and internal sequence of β4 in the reference mouse genome sequence were mapped to the β4 sequence, and a recruitment plot was constructed using the R package enveomics in R version 1.3 with the scripts in the Enveomics Collection[62]. A region showing a depth of less than four was defined as a low copy-number region of β4 for haplotype calling.

For haplotype calling, BAM reads were minimally filtered by the PrintReads command using GATK with options '-rf OverclippedRead -NoRequireSCBothEnds -filterTooShort 80 -rf MappingQuality -mmq 20 -rf UnmappedRead -rf NotPrimaryAlignment -rf FailsVendorQualityCheck -rf DuplicateRead -rf BadMate -rf MateSameStrand'. The copy-number of β4 was estimated by the average depth in the low copy-number region divided by the genome coverage. The rounded value of the copy-number was used as an estimated ploidy for haplotype assembly. The estimated ploidy of each strain is summarized in Supplementary Table 3. For SNP calling, the HaplotypeCaller command was used in GATK with option '-ERC GVCF'. All of the gVCF files were merged by the CombineGVCFs command using GATK, and then SNPs and indels were called by the GenotypeGVCFs command using GATK. HapCompass[63] version 0.8.2 was used for haplotype assembly with the estimated ploidy level. FASTA sequences for each haplotype in the low copy-number region of the β4 sequence were constructed by the FastaAlternateReferenceMaker command using GATK. The phylogenetic tree was constructed using sequences aligned to the homologous sequence of *Microcebus murinus* ERV-β4 (AC145758.1) as an outgroup.

**Data manipulation and scripting**. For bioinformatics analysis, the Bash shell of the Cygwin package was used in a 64-bit version of Windows 7. Data were manipulated using custom shell scripts. Some programs were reverse-engineered to correctly work on the Windows system. Java™ SE Development Kit 8, Update 171 was used for JAR packages.

**Reporting summary**. Further information on research design is available in the Nature Research Reporting Summary linked to this article.

## Data availability
The nucleotide sequences of the β4 element in the *a* allele have been deposited in the DDBJ database under the accession number BR001522.

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

## Acknowledgements

We thank K. Yoshimi for valuable advice on genome editing. This work was supported by Grants-in-Aid for Scientific Research (KAKENHI) 15H04289 and 19H03270.

## Author contributions

A.T. identified the structural variations, designed the study, conducted the animal experiments, performed PCR, and sequence analyses, performed the computational analysis, and wrote the manuscript. Y.I. performed microinjection of CRISPR/Cas9 into mouse embryos. T.K. contributed to writing the manuscript, provided financial support, and supervised the project.

## Additional information

**Competing interests:** The authors declare no competing interests.

