## [Peer Review File · Communications Biology]

Reviewers' comments:

Reviewer #1 (Remarks to the Author):

The non-agouti coat colour in mice arising from disruption of the agouti locus by a VL30 endogenous retrovirus is one of the best-known phenotypes associated with a transposable element insertion. Here, Tanave and colleagues perform a detailed analysis of this "a" allele of the agouti locus, and discover that loss of agouti gene function arises not from disruption by the VL30 sequence itself, but from a previously uncharacterised beta-4 retrovirus insertion nested within the VL30 sequence. They convincingly demonstrate that the presence of the beta-4 retrovirus results in aberrant splicing of the agouti gene. Critically, they use CRISPR/Cas9 mediated deletion to generate agouti alleles lacking the beta-4 retrovirus insertion and lacking the nested VL30/beta-4 sequence altogether, and show that excision of the beta-4 retrovirus insertion (while leaving the VL-30 intact) is sufficient to restore the agouti phenotype and normal splicing of the agouti gene, although in the beta-4 deletion allele some aberrant splice products remain. Phylogenetic analysis and characterisation of the agouti locus in inbred mouse strains confirms the VL30/beta-4 retrovirus nested sequence as responsible for the non-agouti phenotype, and demonstrates the origin of the non-agouti "a" allele via stepwise retrotransposition events in the lineage from East Asian mice to Japanese Fancy mice.

This work is thorough, well executed, and convincing. The use of CRISPR/Cas9 mediated deletion to demonstrate that the beta-4 retrovirus insertion is truly responsible for the non-agouti phenotype elevates this paper significantly. This work updates our understanding of this classical example of a retroelement-mediated phenotype, exposing previously-unappreciated layers of nuance. I believe this work will be well-cited and will make a significant impact in the fields of mobile DNA biology and mammalian genomics. I have no significant criticisms or suggestions for further revision.

Reviewer #2 (Remarks to the Author):

This manuscript by Tanave, Imai, and Koide describes the characterization of the mutation in many strains of mice exhibiting the nonagouti phenotype, including, C57BL/6J. This is a follow-up report from a manuscript by another group of investigators (Bultman et al. 1994) which first described that the nonagouti, a, mutation was caused by the nested insertional events of an unknown retroviral-like element into a VL30 retroviral element into the intron just upstream from the first coding exon of agouti. In this submitted manuscript the authors demonstrate that the previously unknown element within the VL30 element is completely identical to the MmERV-B4_AL805955 element retroviral element. Additionally, the authors demonstrate that the combination of the nested insertion elements interfere with the normal expression of agouti in dorsal skin by producing abnormally spliced transcripts that incorporate portions of the nested retroviral elements and which terminate within the 3' LTR of the B4 element. Most notably, they demonstrate that when the B4 element alone is excised utilizing CRISPR that the animals revert to an agouti coat color phenotype, even though the agouti mRNA's are abnormal with a portion of the remaining VL30 element still splicing into their 5' UTR. They also demonstrate that excising both nested elements utilizing CRISPR results in an agouti coat color phenotype, which is what would be expected. They provided additional data to suggest that the allele within the common laboratory strains was originally derived from the East Asian musculus group, including JF1.

While this work is well done and the manuscript is well written, my biggest concern is that they fail to address why their results appear to conflict with the original Bultman et al (1994) report which demonstrated that 11 independently derived reversion mutations to the black-and-tan (a^t) phenotype that arose in the C57BL/6J production colony over a two year period contained the VL30

element. Based on the Bultman et al (1994) result, one would have expected that the CRISPR deletion of only the B4 element would have generated mice exhibiting the black-and-tan phenotype. Is it possible that in the reversion mutations reported by Bultman et al (1994) that the remaining solo LTR from the B4 element within the VL30 element interferes with the production of normal agouti mRNA? Are there potentially splicing or transcription termination elements within the remaining solo B4 LTR that could interfere with normal transcription?

It would have also been nice if they had classified the mice from their CRISPR experiments as either A (agouti) or A^w (white bellied agouti). It wasn't clear from the photographs in Supplementary Fig. 4.

In the original Bultman et al. (1994) paper, an upstream agouti promoter was described that produced transcripts (called form II transcripts) specifically over the ventral skin of the animal. If the goal of this paper is to show how the unique configuration of the a allele of agouti interferes with the normal expression of the gene, it would be useful to know if/how the nested retroviral elements interfere with the expression of these form II transcripts. Specifically, it would be useful to know if the VL30 element in the CRISPR generated B4-del mutation still interferes with the expression of the form II transcripts.

In the experiments where they studied the expression of the agouti gene using RNAseq, they failed to address the issue that the a allele is not a complete null mutation. According to the Silver's reference, individual hairs in a/a animals are almost exclusively pigmented with eumelanin, but the hairs originating on and behind the ears as well as the hairs around the genital papilla and mammae, are yellow (at least in part). What is likely the mechanisms that regulates agouti in the skin supporting the hairs behind the ears and in the genital papilla and mammae? Clearly in these parts of the animal the nested retroviral insertions are not blocking the expression of agouti in the same way as they describe in the dorsal skin.

Overall, I thought the work was most interesting from the standpoint that it shows the complexity associated with how retroviral elements can impact the expression of genes. Mammalian genomes are full of repetitive elements, many of which are retroviral elements with promoters, enhancers and potential splice sites, and I predict that there will be increasing interest in knowing how these elements could interfere with the normal of expression of genes within close proximity. The case of the a allele is an interesting illustration of how this happens. I suggest that the authors highlight this feature of their work, which may be of interest to a greater audience.

Finally, there are a number of other minor issues that the authors should address:

1. It wasn't clear that the whole description of how portions of the B4 element were found to be associated with a number of other genes, e.g. Ednrb, Fuca1, Gimap3, Nyx, etc. was important to finally reach the conclusion that the unknown retroviral element is actually B4, especially given that they eventually found that the "unknown element is completely identical to that of the previously identified MmERV-B4-AL8055955." Actually, I found this to be distracting. This could be deleted without affecting the impact of the paper.
2. They suggest that the agouti locus allele carried by MSM be referred to as A^{mol}. Did they find that the other lines of mice that carry just the VL30 also express the altered form of mRNA? If so, should these other lines also have the same designation?
3. On Fig. 3a, it was unclear what the RNAseq read peaks represent in the shaded gray box (which are absent in the C3H mouse). Any sense of what the huge read count peak is that is under the B4 LTR (which doesn't connect to any of the junctional reads)—are they missing an aberrant slice form in their analyses? What do they attribute the 11 reads that connect the coding exons 3 and 4 in the B6 skin samples? Also, it is unclear how transcripts 1 and 2 differ from each other.
4. In Fig. 3b, which primers were used for the exon 2-3 PCR.

Reviewer #1 (Remarks to the Author):

Comment: *The non-agouti coat colour in mice arising from disruption of the agouti locus by a VL30 endogenous retrovirus is one of the best-known phenotypes associated with a transposable element insertion. Here, Tanave and colleagues perform a detailed analysis of this "a" allele of the agouti locus, and discover that loss of agouti gene function arises not from disruption by the VL30 sequence itself, but from a previously uncharacterised beta-4 retrovirus insertion nested within the VL30 sequence. They convincingly demonstrate that the presence of the beta-4 retrovirus results in aberrant splicing of the agouti gene. Critically, they use CRISPR/Cas9 mediated deletion to generate agouti alleles lacking the beta-4 retrovirus insertion and lacking the nested VL30/beta-4 sequence altogether, and show that excision of the beta-4 retrovirus insertion (while leaving the VL-30 intact) is sufficient to restore the agouti phenotype and normal splicing of the agouti gene, although in the beta-4 deletion allele some aberrant splice products remain. Phylogenetic analysis and characterisation of the agouti locus in inbred mouse strains confirms the VL30/beta-4 retrovirus nested sequence as responsible for the non-agouti phenotype, and demonstrates the origin of the non-agouti "a" allele via stepwise retrotransposition events in the lineage from East Asian mice to Japanese Fancy mice.*

This work is thorough, well executed, and convincing. The use of CRISPR/Cas9 mediated deletion to demonstrate that the beta-4 retrovirus insertion is truly responsible for the non-agouti phenotype elevates this paper significantly. This work updates our understanding of this classical example of a retroelement-mediated phenotype, exposing previously-unappreciated layers of nuance. I believe this work will be well-cited and will make a significant impact in the fields of mobile DNA biology and mammalian genomics. I have no significant criticisms or suggestions for further revision.

Response: We appreciate this reviewer's encouraging comments.

Reviewer #2 (Remarks to the Author):

Comment: *This manuscript by Tanave, Imai, and Koide describes the characterization of the mutation in many strains of mice exhibiting the nonagouti phenotype, including, C57BL/6J. This is a follow-up report from a manuscript by another group of investigators (Bultman et al. 1994) which first described that the nonagouti, a, mutation was caused by the nested insertional events of an unknown retroviral-like element into a VL30 retroviral element into the intron just upstream from the first coding exon of agouti. In this submitted manuscript the authors demonstrate that the previously*

unknown element within the VL30 element is completely identical to the MmERV-B4_AL805955 element retroviral element. Additionally, the authors demonstrate that the combination of the nested insertion elements interfere with the normal expression of agouti in dorsal skin by producing abnormally spliced transcripts that incorporate portions of the nested retroviral elements and which terminate within the 3' LTR of the B4 element. Most notably, they demonstrate that when the B4 element alone is excised utilizing CRISPR that the animals revert to an agouti coat color phenotype, even though the agouti mRNA's are abnormal with a portion of the remaining VL30 element still splicing into their 5' UTR. They also demonstrate that excising both nested elements utilizing CRISPR results in an agouti coat color phenotype, which is what would be expected. They provided additional data to suggest that the a allele within the common laboratory strains was originally derived from the East Asian musculus group, including JF1.

*While this work is well done and the manuscript is well written, my biggest concern is that they fail to address why their results appear to conflict with the original Bultman et al (1994) report which demonstrated that 11 independently derived reversion mutations to the black-and-tan (*a'*) phenotype that arose in the C57BL/6J production colony over a two year period contained the VL30 element. Based on the Bultman et al (1994) result, one would have expected that the CRISPR deletion of only the B4 element would have generated mice exhibiting the black-and-tan phenotype. Is it possible that in the reversion mutations reported by Bultman et al (1994) that the remaining solo LTR from the B4 element within the VL30 element interferes with the production of normal agouti mRNA? Are there potentially splicing or transcription termination elements within the remaining solo B4 LTR that could interfere with normal transcription?*

Response: We appreciate the reviewer's comment on this important point. Indeed, the reviewer is correct. The solo LTR from $\beta 4$ is sufficient to change the coat color from nonagouti to black-and-tan (*a'*). In our genome-editing experiment using CRISPR/Cas9, we also generated a mutant mouse that carries a single LTR of $\beta 4$ in the VL30 sequence of the *nonagouti* allele, resulting in a black-and-tan (*a'*) allele. The solo $\beta 4$ LTR possesses a splice acceptor site, as shown in Fig. 3c, which is spliced in *a'* mice. Although the mechanism for transcription termination of the *agouti* transcript in the *a'* mice remain unclear, our results showed that the solo $\beta 4$ LTR interferes with normal transcription in dorsal skin. Because this is an important point for understanding the effect of $\beta 4$ in the regulation of coat color in mice, we have added these results to the new Fig. 4 and the revised manuscript (pp 2, lines 39-41; pp 8-9, lines 192-219; Fig. 4; Supplementary Fig. 4, 5; pp 13-14, lines 315-330; pp 19, lines 444-446; Supplementary Table 2).

Comment: *It would have also been nice if they had classified the mice from their CRISPR*

experiments as either A (agouti) or A^w (white bellied agouti). It wasn't clear from the photographs in Supplementary Fig. 4.

Response: The bellies of A^w mice are white, cream, or tan (yellow) depending on the genetic background. Therefore, the coat color of the revertant of the *nonagouti* (*a*) allele is frequently referred to as A^w (white-bellied agouti) but sometimes as A^w (light-bellied agouti) or A^L (light-bellied agouti). Moreover, the color of a spontaneous revertant of *nonagouti* that occurred in the C57BL/6J background, A^{w-J}, is also known as “light- or white-bellied agouti coat”, as described in MM Dickie’s (1969) report. This indicates that the nomenclature of A^w is variable or unclear. For these reasons, we avoided designating the color as A^w and have just noted the cream-colored ventrum in the description of Supplementary Fig. 3. We adjusted the contrast of the photos of the mice in Supplementary Fig. 3 to show the color more clearly.

Comment: *In the original Bultman et al. (1994) paper, an upstream agouti promoter was described that produced transcripts (called form II transcripts) specifically over the ventral skin of the animal. If the goal of this paper is to show how the unique configuration of the a allele of agouti interferes with the normal expression of the gene, it would be useful to know if/how the nested retroviral elements interfere with the expression of these form II transcripts. Specifically, it would be useful to know if the VL30 element in the CRISPR generated B4-del mutation still interferes with the expression of the form II transcripts.*

Response: We appreciate this advice. We added RT-PCR results that indicate that the form II transcripts were also interrupted by abnormal splicing of the retroviral elements (pp 9, lines 202-218, Fig. 4, Supplementary Fig. 5; Supplementary Table 2).

Comment: *In the experiments where they studied the expression of the agouti gene using RNAseq, they failed to address the issue that the a allele is not a complete null mutation. According to the Silver’s reference, individual hairs in a/a animals are almost exclusively pigmented with eumelanin, but the hairs originating on and behind the ears as well as the hairs around the genital papilla and mammae, are yellow (at least in part). What is likely the mechanisms that regulates agouti in the skin supporting the hairs behind the ears and in the genital papilla and mammae? Clearly in these parts of the animal the nested retroviral insertions are not blocking the expression of agouti in the same way as they describe in the dorsal skin.*

Response: This question remains under study. As suggested by Vrieling et al.’s (1994) report, follicular melanocytes in certain areas such as behind the ears and around the genital papillae and

mammae might be especially sensitive, or another regulatory mechanism that acts at a location other than the known transcription start site (TSS) may produce agouti expression in those areas. We suspect the latter possibility, as Fontanesi et al. (2009) reported that expression from an alternative TSS upstream of the *agouti* exon 2 in the dorsal skin of rabbits. However, we do not have any data to distinguish between these possibilities. Therefore, we believe that detailed further study is needed to clarify these points and thus it is better to avoid discussion in the current paper.

Comment: *Overall, I thought the work was most interesting from the standpoint that it shows the complexity associated with how retroviral elements can impact the expression of genes. Mammalian genomes are full of repetitive elements, many of which are retroviral elements with promoters, enhancers and potential splice sites, and I predict that there will be increasing interest in knowing how these elements could interfere with the normal of expression of genes within close proximity. The case of the a allele is an interesting illustration of how this happens. I suggest that the authors highlight this feature of their work, which may be of interest to a greater audience.*

Response: We appreciate this important comment. We have added a detailed explanation of how the $\beta 4$ retroviral element interferes with normal *agouti* expression (pp 13, lines 296-313).

Comment: *1. It wasn't clear that the whole description of how portions of the B4 element were found to be associated with a number of other genes, e.g. Ednrb, Fucal, Gimap3, Nyx, etc. was important to finally reach the conclusion that the unknown retroviral element is actually B4, especially given that they eventually found that the "unknown element is completely identical to that of the previously identified MmERV-B4-AL8055955." Actually, I found this to be distracting. This could be deleted without affecting the impact of the paper.*

Response: We mostly agree with the reviewer's comment, but we think that the comparison between the unknown retroviral element in VL30 and the inserted element in *Ednrb*^s and other genes is informative to the discussion of how these related elements can interfere with normal transcript expression. Therefore, we have retained most of the description of the other genes but have removed the previous Supplementary Fig. 1 from the revised manuscript.

Comment: *2. They suggest that the agouti locus allele carried by MSM be referred to as A^{mol}. Did they find that the other lines of mice that carry just the VL30 also express the altered form of mRNA? If so, should these other lines also have the same designation?*

Response: We found that the SWN, KJR, and CHD strains have the same *agouti* haplotype as MSM and the VL30 insertion in the *agouti* locus. However, we have not analyzed the actual RNA expression in SWN, KJR, and CHD, and we are unable to show data confirming the expression of the *agouti* gene in these strains. Given that the aim of our study was not to investigate the *agouti* allele among strains, we would like to leave this for future work.

Comment: 3. On Fig. 3a, it was unclear what the RNAseq read peaks represent in the shaded gray box (which are absent in the C3H mouse). Any sense of what the huge read count peak is that is under the B4 LTR (which doesn't connect to any of the junctional reads)? are they missing an aberrant splice form in their analyses? What do they attribute the 11 reads that connect the coding exons 3 and 4 in the B6 skin samples? Also, it is unclear how transcripts 1 and 2 differ from each other.

Response: In Fig. 3a, the shaded box region represents VL30, which does not exist in the C3H mouse. The RNA-seq data from both B6 and C3H were mapped onto the B6 genome. In C3H, expression peaks were seen at VL30 on the B6 reference genome. Given that VL30 is present in many copies in the mouse genome, the short reads of expressed sequences from VL30 in C3H could not be distinguished from those of other VL30 copies and thus mapped to this region. We have mentioned this point in the manuscript (pp 8, lines 176-182; pp 32, and in the figure legend for Fig. 3).

Moreover, a huge read count peak at the β 4 LTR was not connected to any another peak. In the analysis, we were unable to distinguish between short reads mapping to the B4 LTR at either the 5' or 3' ends. The β 4 LTR is a repetitive sequence in the mouse genome, as shown in Supplementary Fig. 9. Thus, it was difficult to map short reads to a single β 4 LTR in the mouse genome. The RNA-seq read peak under the β 4 LTR is not connected to any another sequence, suggesting that these sequences could be mapped to other β 4 LTRs in different genomic regions. We have also mentioned this point in the manuscript (pp 8, lines 170-172).

We believe that the 11 reads that connect the coding exons 3 and 4 in the B6 skin sample reflect transcripts initiated upstream of exon 3. In fact, some cDNAs initiated upstream of exon 3 have been reported in nucleotide databases, and CAGE-based TSS analysis showed a strong signal around the region upstream of exon 3. Thus, these unusual reads indicate the presence of non-functional RNAs from the *agouti* gene. However, we do not have supporting data to confirm this, so we decided not to mention this point in the current paper.

Transcripts 1 and 2 differ in their splice acceptor sites adjacent to the VL30 LTR sequence. We have revised Fig. 3c and indicated the position of the splice acceptor site with asterisks (Figure legends).

Comment: 4. In Fig. 3b, which primers were used for the exon 2-3 PCR.

Response: We have revised the figure to show which primers were used for the exon 2–3 PCR.

REVIEWERS' COMMENTS:

Reviewer #1 (Remarks to the Author):

I have no further concerns for the authors.

Reviewer #2 (Remarks to the Author):

The authors have done an excellent job addressing all of my original comments. The only point that was not clear was how they generated the animals $\beta 4(\text{direct-rec})$ mutant animals. In methods they indicate that "The $\beta 4(\text{direct-rec})$ mice were obtained during the above CRISPR-mediated deletion of $\beta 4$ in the a allele. Direct-repeat recombination was confirmed by genomic PCR (Supplementary Fig. 4b) and direct sequencing of the PCR product (Supplementary Fig. 4a)." Did the $\beta 4(\text{direct-rec})$ mutation arise by spontaneous homologous recombination, or do they use CRISPR to facilitate the generation of the mutant?

Reviewer #1 (Remarks to the Author):

Comment:

I have no further concerns for the authors.

Response: We thank the reviewer for reviewing our manuscript.

Reviewer #2 (Remarks to the Author):

Comment: *The authors have done an excellent job addressing all of my original comments. The only point that was not clear was how they generated the animals $\beta 4(\text{direct-rec})$ mutant animals. In methods they indicate that "The $\beta 4(\text{direct-rec})$ mice were obtained during the above CRISPR-mediated deletion of $\beta 4$ in the *a* allele. Direct-repeat recombination was confirmed by genomic PCR (Supplementary Fig. 4b) and direct sequencing of the PCR product (Supplementary Fig. 4a)." Did the $\beta 4(\text{direct-rec})$ mutation arise by spontaneous homologous recombination, or do they use CRISPR to facilitate the generation of the mutant?*

Response: We appreciate the reviewer's encouraging comment. We believe that the $\beta 4(\text{direct-rec})$ mutation was facilitated by use of CRISPR but we do not have supportive data as this mutation was occurred spontaneously during the genome editing. We mentioned this point in the Methods (pp 19, lines 449-452).